# Comprehensive Analysis of Purine-Metabolism-Related Gene Signature for Predicting Ovarian Cancer Prognosis, Immune Landscape, and Potential Treatment Options

**DOI:** 10.3390/jpm13050776

**Published:** 2023-04-29

**Authors:** Jingchun Liu, Xiaoyi Zhang, Haoyu Wang, Xiaohu Zuo, Li Hong

**Affiliations:** Department of Obstetrics and Gynecology, Renmin Hospital of Wuhan University, Wuhan 430060, China

**Keywords:** purine metabolism, ovarian cancer, personalized medicine, bioinformatical analysis, drug sensitivity, extensive non-targeted metabolomics

## Abstract

Purine metabolism is an important branch of metabolic reprogramming and has received increasing attention in cancer research. Ovarian cancer is an extremely dangerous gynecologic malignancy for which there are no adequate tools to predict prognostic risk. Here, we identified a prognostic signature consisting of nine genes related to purine metabolism, including ACSM1, CACNA1C, EPHA4, TPM3, PDIA4, JUNB, EXOSC4, TRPM2, and CXCL9. The risk groups defined by the signature are able to distinguish the prognostic risk and the immune landscape of patients. In particular, the risk scores offer promising personalized drug options. By combining risk scores with clinical characteristics, we have created a more detailed composite nomogram that allows for a more complete and individualized prediction of prognosis. In addition, we demonstrated metabolic differences between platinum-resistant and platinum-sensitive ovarian cancer cells. In summary, we have performed the first comprehensive analysis of genes related to purine metabolism in ovarian cancer patients and created a feasible prognostic signature that will aid in risk prediction and support personalized medicine.

## 1. Introduction

Ovarian cancer (OC) is the seventh most common cancer in women. As a result of ineffective early detection tools, most OC patients are found at an advanced stage [1,2]. Although the overall five-year relative survival rate was approximately 49%, it was only 29% for patients with late-stage OC [3]. Despite the clinical achievements of progressive treatment modalities, high mortality rates persist [4,5,6]. The classic OC markers CA125 and HE4 provide rough diagnostic capability of the disease, but the false positivity and lack of prognostic predictive power cannot be ignored [7,8]. Other emerging biomarkers are being widely explored and are yet to be tested on a large scale [9]. Therefore, there is an urgent need to identify markers or tools for OC diagnosis, risk prediction, and prognostic assessment.

Metabolic reprogramming is an essential hallmark of tumors. The aberrant purine metabolism in tumors has gained increasing attention, mainly for supporting cell proliferation by providing necessary substrates for nucleotide synthesis [10]. In addition, the interaction of purine metabolism with mitochondria has been gradually revealed [11]. Multiple studies have reported the intervention of purine metabolism on the progression of human solid tumors, including promotion of proliferation, invasion, metastasis, stemness, and therapeutic resistance [12,13]. Purine-metabolism-related genes (PRGs) are a broad group of genes involved in the grand process of purine metabolism and have potentially important effects on metabolic activity. The mRNA levels of PRG can be easily and quickly obtained. Clinical testing based on PRGs may make achievable assessments of future risk for oncology patients.

Although some studies have developed prognostic models related to PRGs, the evidence is insufficient for OC and can be further optimized. Therefore, in our current study, we reconstructed a PRG-related signature applicable to the prognosis prediction of OC patients based on gene expression arrays from public databases, including Genotype-Tissue Expression (GTEx), The Cancer Genome Atlas (TCGA), and Gene Expression Omnibus (GEO). The risk scores and subgroups provided by the signature have the capability to map the immune landscape of the patient and indicate a personalized therapeutic regimen. We also integrated the analysis of PRG signatures and clinical characteristics to create a composite nomogram for improving the applicability of the model. In addition, extensive non-targeted metabolomics revealed differences between platinum-sensitive and platinum-resistant OC cells. These findings may provide prognostic predictions and alternative combinations of drugs for future OC patients and contribute to the development of future metabolism-related targeted drugs.

## 2. Materials and Methods

### 2.1. Datasets

Gene expression data (FPKM format), mutation data, clinical status, and characteristics of OC patients in the original cohort were obtained from TCGA. To avoid oversight, data from all 427 OC patients were used for the gene expression files. Mutation data, clinical status, and characteristics were obtained from 377 OC patients with complete clinical information. Data from 88 normal ovarian tissues, which served as controls, were obtained from GTEx. Data for the external validation cohort were downloaded from the GEO database (ID: GSE53963). All expression values were log2-transformed. Purine-metabolism-related genes (PRGs) were obtained from GeneCards database (https://www.genecards.org/ (accessed on 30 May 2022)). The criteria for inclusion were protein-coding category and a relevance score of at least 1.

### 2.2. Identification of Candidate Genes

Differentially expressed genes (DEGs) were obtained based on the R package “limma” and the Benjamini–Hochberg method. Screening criteria were |log2FC| > 1 and adjusted *p* value > 0.05. The intersection of purine-metabolism-related protein-encoding genes and DEGs gave differentially expressed PRGs. Univariate Cox regression analysis was performed to identify genes associated with prognosis (*p* value < 0.05). Multiple intersections were then used to identify the PRGs for further analysis.

### 2.3. Gene Function Enrichment Analysis

Gene function enrichment analysis was performed using the R package “clusterProfiler”. Enrichment was based on the Kyoto Encyclopedia of Genes and Genomes (KEGG) comprehensive database and Gene Ontology (GO) annotation with the screening criteria of adjusted *p* value < 0.05 and q value < 0.2. Construction of the protein–protein interaction (PPI) network, molecular complex detection (MCODE) components, and tissue specificity analysis obtained from PaGenBase were performed by Metascape Online (https://metascape.org/gp/index.html (accessed on 30 May 2022)).

### 2.4. Construction and Validation of the PRG-Based Prognostic Signature

PRG-based prognostic signatures were constructed using TCGA as the training cohort and GEO (GSE53963) as the validation cohort. The tenfold cross-validated LASSO regression analysis was first performed on the TCGA cohort to narrow down the candidate genes for the signature genes, and 21 candidate gene variables with the minimum penalty parameter (λ) were identified. Multivariate Cox regression identified nine optimal prognostic genes, and risk score formulas were constructed based on the corresponding coefficients. High- and low-risk subgroups of the two cohorts were distinguished based on the median risk score. Survival status, OS time, and gene expression levels were implemented using the R packages “stats”, “survival”, and “heatmap”. Kaplan–Meier curves were used to characterize OS. Receiver-operating characteristic (ROC) curves were generated to confirm the effectiveness of prediction.

### 2.5. Mutation Landscape and Subgroup Survival Analysis

The mutational landscape of the TCGA cohort was extracted and described using the R package “maftools”. Kaplan–Meier curves characterized survival analysis by mutation subgroups and clinical subtypes. Correspondence between risk groups, clinical characteristics, and major mutation subtypes was depicted in a Sankey diagram.

### 2.6. Immune Infiltration and Checkpoint Analysis

CIBERSORT (https://cibersortx.stanford.edu/ (accessed on 8 June 2022)), a deconvolution algorithm based on transcriptomic data for each sample, was run to estimate the fraction of 22 immune cells. Meanwhile, transcriptomic data were also used to assess immune checkpoint expression in different subgroups to evaluate the benefits of immune checkpoint blockade therapies.

### 2.7. Drug Sensitivity Analysis

Gene expression and drug half-inhibitory concentration (IC50) data for OC cell lines were provided by CellMiner [14] (https://discover.nci.nih.gov/cellminer/ (accessed on 8 June 2022)) and GDSCv2 (https://www.cancerrxgene.org/ (accessed on 8 June 2022)). For the CellMiner database, we profiled drugs that had been tested in clinical trials and approved by the FDA with at least three quality-controlled data. To avoid neglecting potential drugs, the same analysis was performed in GDSCv2.

### 2.8. Prognostic Independence Assessment and Construction of Nomogram

Univariate and multivariate Cox regression analyses of clinical characteristics and risk scores were performed to identify independent prognostic factors. In addition, the R package “rms” and Cox regression analysis were used to create a novel nomogram with complete clinical characteristics and risk scores to allow more comprehensive and personalized prediction of prognosis. Calibration curve and decision curve analyses were used to assess the degree of fit or net clinical benefit.

### 2.9. Cell Lines and Cell Culture

Human normal ovary epithelial cells IOSE, human OC cells A2780, SKOV3, and cisplatin-resistant cells SKOV3/DDP were obtained from the China Center for Type Culture Collection (Wuhan, China). All cells were cultured in RPMI 1640 (Gibco, Billings, MT, USA) containing 10% fetal bovine serum (Yeasen, Shanghai, China). Regular medium changes and passages were performed according to the cell status.

### 2.10. RNA Extraction and Quantitative RT-PCR

According to the manufacturer’s instructions, MolPure Cell/Tissue Total RNA Kit (Yeasen) and Hifair II 1st Strand cDNA Synthesis Kit (Yeasen) were used to extract total RNA and reverse transcription into cDNA. Then Hieff qPCR SYBR Green Master Mix (Yeasen) was added to premix the primers and samples for cDNA amplification. β-Actin was used as a normalized control. All data were repeated at least three times.

The primer sequences used were as follows: ACSM1, AAAGGAGAAGGAGGGCAAGAGAGG and TTCAACAGGATGGTCGCAGGAATG; CACNA1C, GAAGAGGATGAGGAGGAGCCAGAG and CAGCAGGTCCAGGATGTTGAAGTAG; CXCL9, TTTGGCTGACCTGTTTCTCCC and GGTCGCTGTTCCTGCATCAG; EPHA4, ACCAACCAAGCAGCACCATCATC and CCCGAGACAGAGACCAGAAGGAC; EXOSC4, CTCTTGTCGGACCAGGGCTACC and GCTGTGTGAGGATGGCTGCTTC; JUNB, GCCACCTCCCGTTTACACCAAC and CCTTGAGCGTCTTCACCTTGTCC; PDIA4, TCAGCAAGCGTTCTCCTCCAATTC and CTCTCAGGTTGTTAGCGGCATCC; TPM3, CTTGGAACGCACAGAGGAACGAG and CAGCAAACTCAGCACGGGTCTC; TRPM2, TGGCGGAGGAGTATGAGCACAG and CAGAGGAAGGCGAAGTAGGAGAGG.

### 2.11. Immunohistochemistry

Immunohistochemical images were provided by the Human Protein Atlas (HPA) database (https://www.proteinatlas.org/ (accessed on 30 May 2022)). Representative images of normal ovarian tissue and ovarian malignancies were selected.

### 2.12. Comprehensive Nontarget Metabolomic Analysis

Briefly, metabolites were extracted using an aqueous 80% methanol solution in cleaned cells of appropriate growth density and then centrifuged. The sample supernatants were analyzed by a high performance liquid chromatography–electrospray tandem mass spectrometry (HPLC-ESI-MS) system. The relative amounts and their attribution to metabolic pathways of representative metabolites in the samples are presented in a heat map. Specific data were log-transformed and mean-centered. Differential metabolites were identified based on VIP ≥ 1 and |Log2FC| ≥ 1. The identified metabolites were annotated and mapped using the KEGG compound database, and *p* values from hypergeometric tests were used to identify significantly enriched pathways.

### 2.13. Statistical Analysis

In addition to the above statistical methods, correlations were described using Pearson correlation analysis. Kaplan–Meier curve analysis was performed using the log-rank test; the horizontal axis represents the time measured in days. Statistical differences in immune infiltration, drug sensitivity analysis, and quantitative RT-PCR were determined by Student’s *t*-test, with a *p* value < 0.05 considered significant. All statistical analyses and image plots were generated using R software (v4.1.0) or GraphPad Prism 8.

## 3. Results

### 3.1. Identification of Dysregulated PRGs

We compared DEGs in 88 normal ovarian tissues from GTEx and 377 ovarian tumor tissues from TCGA (Figure 1A). After matching the list from the GeneCards database, a total of 978 PRGs were finally identified (Figure 1B). Univariate Cox regression analysis further revealed 114 genes identified to be associated with prognosis (Figure 1C & Appendix A). To explore the potential mechanisms of these genes, functional analysis was performed. GO analysis suggested that these dysregulated PRGs with prognostic significance were enriched in response to mechanical stimulus, purine-containing compound, and organophosphorus (Figure 1D). Meanwhile, KEGG pathway analysis demonstrated a significant enrichment in human immunodeficiency virus 1 infection, GnRH signaling pathway, and Epstein–Barr virus infection. These results prompted us to further investigate the relationship between purine metabolism gene sets, tumor prognosis, and the immune microenvironment. In addition, a PPI network constructed based on metascape identified the MCODE components of these dysregulated purine metabolism genes that may be key regulatory combinations (Figure 1E,F). Data from PaGenBase also supported the specificity of these genes in thyroid, ovary, and spleen tissue (Figure 1G).

### 3.2. Construction and Validation of a PRG-Based Prognostic Signature

We used the LASSO regression method to first test the genes used to generate the prognostic signature. The most stable model was found at a minimum value of the penalty parameter (λ = 0.067) (Figure 2A,B). We then selected 21 candidate gene variables under this optimal condition for further analysis. ACSM1, CACNA1C, EPHA4, TPM3, PDIA4, JUNB, EXOSC4, TRPM2, and CXCL9 were identified as independent prognostic factors by multifactorial Cox regression (Table 1). Based on the baseline coefficients, we constructed a prognostic feature based on nine PRGs (Figure 2C,D).

The median risk score divided patients into high-risk groups and low-risk groups. The distribution of OS times showed that higher risk scores were associated with a worse prognosis (Figure 2E). Kaplan–Meier analysis showed a significant separation of OS between high- and low-risk patients (Figure 2F). We plotted ROC curves to evaluate the sensitivity and specificity of the prognostic signature and found that the area under the curve (AUC) was 0.768 at 1 year, 0.702 at 4 years, and 0.758 at 6 years, demonstrating good predictive validity of the signature (Figure 2G).

To further confirm the predictive power of the signature, we included 174 patients from GSE53963 in the validation cohort. Higher survival and lower mortality were observed in the low-risk group (Figure 2H,I). In addition, we assessed the external predictive power of the prognostic signature. The AUCs were 0.722 (1 year), 0.603 (4 years), and 0.709 (6 years), demonstrating considerable predictive validity and robustness of the signature in external validation (Figure 2J).

### 3.3. Survival Prognosis of Different Mutation Subtypes and Different Clinical Subgroups

To investigate the association of our signature with common mutations in OC, the TCGA cohort was analyzed for different single nucleotide variants. The mutational landscape was dominated by TP53 and TTN mutations in both the high- and low-risk groups (Figure 3A). Therefore, we generated Kaplan–Meier curves for these patients according to whether a TP53 mutation or a TTN mutation was present. Remarkably, the prognosis of the high-risk group was worse than that of the low-risk group, regardless of the presence of a TP53 mutation or TTN mutation (Figure 3B,C). Notably, the hazard ratio(HR) showed a large difference of more than twofold in the population with and without TTN mutation. This finding reflects the more important diagnostic validity of the signature in the TTN mutant population.

However, how our signature risk scores interact with clinical characteristics and prognosis is not yet well understood. Therefore, as shown in Figure 3D, we combined the risk score with clinical characteristics, including age, TP53, or TTN mutation, FIGO stage, and tumor grade. The results showed that high-risk patients had disturbingly low survival rates in both the young and old groups (Figure 3E,F). We also found that in a clinical high-risk population, such as FIGO stage III or grade 3&4, the median time to OS was significantly longer in both low-risk populations identified by the prognostic signature (Figure 3G,H).

### 3.4. Immune Cell Landscape and Immune Checkpoints

The results of the gene function enrichment analysis prompted us to focus on the components of immune infiltration. Therefore, we examined the proportion of immune cells in the TCGA cohort according to the CIBERSORT algorithm. As shown in Figure 4A, T cells CD4 and macrophages accounted for a large proportion of the infiltrating immune cells. In addition, we found that the fraction of macrophages M1 was significantly lower in the high-risk group than in the low-risk group (Figure 4B). In the GEO cohort, the immune cell landscape was dominated by the T-cell CD8, B cell memory, and T cell CD4 naïve phenotypes (Figure 4C). The trend in macrophage M1 was similar to TCGA, but did not show statistical differences (Figure 4D). Another finding was that the high-risk group in the GEO cohort contained more resting dendritic cells and activated mast cells.

Given the advances in immune checkpoint blockade therapies [15], we examined the expression of classical immune checkpoints and found that patients in the low-risk group had a relatively higher expression of TIGIT and IDO1 in the TCGA cohort (Figure 4E). However, we noticed that the significant change in expression of IDO1 was consistent in both cohorts, and not that of TIGIT (Figure 4F). These results suggest that the potential sensitivity to IDO1 inhibitors may vary by risk group.

### 3.5. Drug Sensitivity Analysis

Next, based on the CellMiner database, we analyzed the sensitivity of 42 FDA-approved and clinically-tested drugs in OC cell lines. The sensitivity of different risk groups to drugs is shown in Figure 5A. Considering the neglect of promising drugs by dichotomous classification, we focused on their risk scores rather than risk groups and found a strong positive correlation between risk scores and vinblastine IC50 and a negative correlation with bleomycin IC50 (Figure 5B). To avoid information bias from a single database, we performed additional analysis in GDSCv2. Trends in drug sensitivity identified by risk grouping were somewhat different from those of CellMiner (Figure 5C). For example, in data from GDSCv2, differences in vinblastine between risk subgroups were significantly attenuated. In particular, a negative correlation was shown between risk score and dasatinib IC50 (Figure 5D). However, this correlation does not seem to be strong enough and is contrary to the data from CellMiner. Considering the amount of data included and the credibility, we used the data from CellMiner as the main source. Overall, the prognostic profile of PRG indicates the potential of these agents. In particular, bleomycin may be helpful in high-risk patients, whereas vinblastine would be more appropriate in low-risk patients.

### 3.6. Construction of a Comprehensive Nomogram

We performed Cox regression analysis to enhance the clinical usability of our signature based on patients’ risk scores and clinical characteristics such as age, FIGO stage, and tumor grade. Univariate Cox regression analysis showed that age, FIGO stage, and risk score were significantly associated with OS (Figure 6A). However, only the risk score was an independent prognostic factor in multivariate Cox regression analysis (Figure 6B). Moreover, decision curve analysis plots exhibited a good net clinical benefit for our prognostic signature with better risk scores than other clinical characteristics (Figure 6C).

Further, to personalize the prognosis of OC patients, a novel nomogram was created. The nomogram includes risk scores and clinical characteristics, such as age, FIGO stage, and tumor grade, for greater clinical utility. In brief, a specific point is assigned to the level of each influencing factor, and the total points obtained by summing the individual points predict OS at 1, 4, and 6 years (Figure 6D). The overall c-index for the nomogram was 0.669 (95% CI = 0.649–0.690). Calibration curves reflected a good consistency between predicted and observed outcomes for 1-, 4-, and 6-year OS in the TCGA cohort (Figure 6E).

### 3.7. Expression Validation of Nine PRGs and Cellular Metabolome Analysis

We checked the relative mRNA levels of nine PRGs by quantitative RT-PCR in IOSE, A2780, SKOV3, and SKOV3/DDP (Figure 7A). Overall, the mRNA levels of CXCL9, EXOSC4, JUNB, PDIA4, and TRPM2 appear to be conserved in OC cell lines. The expression levels of ACSM1, CACNA1C, EPHA4, and TPM3 appear to be adjusted according to cell line type. Risk scores were calculated for three OC cell lines (Figure 7B). Of note, SKOV3/DDP had a much higher risk score than the parental cell line SKOV3. This suggests that prognostic signatures based on PRGs can detect malignant progression from OC, as drug resistance is an important manifestation of malignant progression. To further determine the protein expression levels of these signature genes, we examined the Human Protein Atlas database (Figure 7C). The proteins TPM3, PDIA4, and EXOSC4 were highly expressed in OC compared with normal ovarian tissue. However, protein levels of CACNA1C and JUNB antibody staining were relatively low in OC tissues. The protein expression of ACSM1 and EPHA4 was not detected in either tumor or normal ovarian tissues. In addition, immunohistochemical data of TRPM2 and CXCL9 were not included.

Because there was a dramatic difference in risk scores associated with PRGs between SKOV3 and SKOV3/DDP, we wanted to know whether there was a corresponding dramatic change in purine metabolism between them. Comprehensive nontarget metabolomic analysis suggested that SKOV3/DDP has a greater amount of nucleotides and their metabolites, which is the main attribution for purine metabolism (Figure 7D). KEGG enrichment analysis of specific metabolites indicated that the purine-metabolism pathway was one of the most conspicuously different pathways. This suggests that platinum-resistant OC cells have greater abundance or activity of purine metabolism. This may provide direction for future personalized treatment of platinum-resistant OC.

## 4. Discussion

Metabolic reprogramming has been recognized as a critical marker of tumor development, providing the necessary energy supply and anabolic demand to promote tumor progression and malignant features [16]. As one of the fundamental metabolic processes for tumor growth, purine metabolism plays an important role in cancer progression, as it provides essential substrates for nucleotide biosynthesis and is involved in the assembly of energy carriers or important coenzymes [10]. Excessive purine synthesis promotes chemotherapy resistance [17]. The end product of purine metabolism, uric acid, influences tumorigenesis and progression to varying degrees [18,19], possibly due to inflammation or oxidative stress [20,21]. OC is an extremely dangerous malignancy, and reliable tools to predict prognostic risk are lacking. We have previously summarized the interference of molecules of purine metabolism in OC [22] and suggested that purine metabolism may significantly influence the malignant features of OC. Therefore, focusing on purine metabolism to develop valuable prognostic signatures is of clinical importance for predicting the prognostic risk of OC and developing new personalized treatment options.

In the current study, we examined several PRGs and found that they were enriched in three tissues, the thyroid, ovaries, and spleen, which may indicate similar regulatory mechanisms and the importance of neglected purine metabolism in these tissues. In fact, studies have reported that the thyroid gland and ovaries regulate each other and that thyroid disease is also associated with polycystic ovary syndrome [23,24,25]. However, more evidence is lacking, and investigation of mechanisms and immune regulation will be important. LASSO and multivariate COX regression analysis were used to determine a signature based on nine PRGs that predicted the prognostic risk of patients with OC. In this signature, CACNA1C, EPHA4, JUNB, and TRPM2 were identified as risk factors, whereas ACSM1, TPM3, PDIA4, EXOSC4, and CXCL9 were protective factors. Although none of the nine genes had high coefficients, the clustering of several genes reflected good predictive performance for prognosis and was confirmed in external validation. The mRNA expression levels of nine PRGs were validated in OC cell lines, and the availability of signatures was confirmed. Combining signature-based risk scores and clinical characteristics, we created a complete nomogram to personalize and more completely determine the prognosis of OC patients. By combining signature-based risk scores and clinical characteristics, we created a complete nomogram to personalize and more completely determine the prognosis of OC patients. The existence of nomograms expands the value of prognostic signatures and contributes to future practical applications.

The prognostic significance of some of these genes in OC has been reported in other studies. High expression of CACNA1C was associated with low survival in patients with OC and was independent of some clinical characteristics such as patient age, pathological grade, FIGO stage, tumor location, and venous infiltration [26]. As part of the chemokine superfamily, CXCL9, in collaboration with CXCR3, activates an immune response in the tumor microenvironment to fight cancer [27]. High CXCL9 expression has been reported to be associated with increased OS in OC patients and is an important potential biomarker for measuring the efficacy of immunotherapy [28]. Similarly, low expression of PDIA4 has been reported to be associated with shorter survival in ovarian tumors, especially in drug-resistant tissues [29,30]. Another interesting point is that high EXOSC4 expression promotes malignant features of OC via the Wnt pathway and correlates with higher FIGO stage and pathological grade, although it is a protective factor in our signature [31]. Although the relationship between other genes and OC has not been adequately studied, they have been reported to play a role in other cancers. High EPHA4 levels have been reported to contribute to the initiation of breast cancer [32], facilitate the acquisition of cisplatin-resistant properties in well-differentiated cervical cancer cells [33], and promote the migration and invasiveness of clear cell renal cell carcinoma and pancreatic ductal adenocarcinoma [34,35]. Previous reports have associated high levels of JUNB in circulating tumor cells with poor prognosis [36]. However, other studies have found that high JUNB expression in the tumor microenvironment can strongly inhibit distant tumor metastasis [37].

In fact, TP53 mutations and TTN mutations have become a common process in the progression of many cancers [38,39]. TP53 is often characterized as a DNA-binding transcription factor involved in the response to multiple stress signals in tumors and plays an oncogenic role in tumors [40]. However, mutant TP53 orchestrates emergency response mechanisms in the tumor microenvironment, contributing to continued tumor survival and progression under adverse conditions. TP53 appears to have a very high mutation frequency in OC, especially in high-grade plasmacytotic OC, which represents a poor prognosis [41,42]. TTN is one of the most commonly mutated genes in the pan-cancer cohort and is associated with poorer prognosis in a variety of human solid tumors [43,44,45]. Mutations in TTN, which encodes a large number of rhabdomeric proteins, may interfere with intrinsically disordered protein regions of tumors or contribute to promote chromosomal instability [46]. Despite the lack of corresponding studies in OC, its association with high mortality has been well analyzed [47]. In addition, OC is less friendly to older women than younger women, as evidenced by a higher mortality rate and a greater likelihood of being initially diagnosed with advanced disease [48]. FIGO staging and grade are well known indicators of OC malignancy and are associated with poor prognosis. Our signature showed good discriminatory power in the presence or absence of TP53 or TTN mutations and superior prognostic diagnostic ability especially in patients with TTN mutations. The signature-based high-risk population was identified with worse prognosis in the cohort of young women, the cohort of older women, the FIGO 3 cohort, and the GRADE 3&4 cohort. These results demonstrate the validity and independence of our signature.

To date, the mechanisms between purine metabolism and OC have not been fully investigated. Some enzymes related to purine metabolism appear to overlap with immunomodulatory mechanisms [49,50], and enrichment analysis assigns extensive PRGs to multiple immune-related pathways. Here, we applied the CIBERSORT algorithm to characterize and compare the immune cell landscape of high-risk and low-risk patients. We found that high risk scores were associated with significantly lower levels of macrophage M1 but higher levels of resting dendritic cells and activated mast cells. Indeed, previous studies have shown that M1 macrophages contribute to antitumor immunity [51], and OC patients with a high M1/M2 ratio have a better prognosis [52]. This agrees with our signature. It is well known that dendritic cells have antigen-presenting and T-cell-stimulating functions [53]. I In our signature, the high-risk group had more resting than activated dendritic cells, suggesting poor antitumor function and possibly even immune escape. There was also a warning of more activated mast cells in the high-risk group. Mast cells have been reported to be activated and release pro-angiogenic factors and matrix metalloproteinases that promote tumor progression [54]. In addition, the abundance of stromal mast cells in OC has been shown to be associated with poor prognosis [55]. In conclusion, based on our signature, we have been able to establish a reliable immune landscape of different risk groups.

Furthermore, IDO1 mRNA levels differed significantly between risk groups in immune checkpoint analysis, suggesting a potential clinical application of IDO1 inhibitors. However, it is unclear whether such differences are the result of IDO1 variabilities and whether they necessarily lead to changes in IDO1 protein levels. These unresolved questions provide food for thought about the potential efficacy of IDO1 inhibitors in different subgroups. Recent studies have shown that IDO1 expression strongly affects purine metabolism in OC and pancreatic cancer [56,57]. In addition, an association between high IDO1 levels and chemoresistance and poor survival has been found in OC patients [58]. These findings provide scope for the development of IDO inhibitors for the treatment of OC. Indeed, IDO1 inhibitors have been found to interfere with the kynurenine pathway to suppress malignant features of tumors [59,60] and synergize with radiotherapy as well as other immunotherapies [61,62]. IDO1 inhibitors are currently being tested in clinical trials in a variety of tumors and have promising potential for further investigation [63,64], although encouraging results are not currently available for OC [65,66].

To demonstrate the therapeutic potential of currently known FDA-approved drugs with respect to our prognostic signature, we analyzed the sensitivity of these drugs in OC. However, the data used were from several OC cell lines that were different from the samples previously used for analysis, and it was difficult to obtain statistical differences for in-depth evaluation. Therefore, we focused our attention on comprehensive risk scores rather than risk groups, while avoiding neglecting potential therapeutics based on simple dichotomization. On this basis, bleomycin and vinblastine were identified as potential agents targeting PRG-based prognostic signature. Bleomycin is particularly suitable for patients with high risk scores, while vinblastine is suitable for low-risk patients. However currently, with vinblastine, single use does not seem to be particularly helpful to patients [67]. Homesley et al. suggested that the combination of bleomycin, etoposide, and cisplatin (BEP) is a viable treatment option for mesenchymal ovarian malignancies [68]. Specifically, studies have reported that PVB chemotherapy (including cisplatin, vincristine, and bleomycin) is superior to CAP chemotherapy (including cyclophosphamide, aclarubicin, and cisplatin) in terms of clinical remission [69]. Personalized treatment management of patients based on characteristics is increasingly being promoted [70]. More experimental options and beneficial combinations need to be explored in the future.

It is worth noting that the creation of the nomogram improves the validity of this prognostic signature based on PRGs. In short, the use of the signature alone only captures the differentiation of risk subgroups and the level of risk scores. Analysis of our prognostic signature together with clinical characteristics allows more accurate prediction of multiyear patient survival. This is a reflection of the development of personalized medicine.

We evaluated the risk scores in different cell lines and found that the risk scores were unstable. However, since these scores were derived from cell line data and not from human OC, we do not consider the significance of comparisons between different species of cell lines to be sufficient. SKOV3/DDP is the platinum-resistant strain derived from SKOV3, so we focused on the differences between the two. An interesting finding was that the risk score was higher in SKOV3/DDP than in SKOV3. To this end, we performed a comprehensive nontargeted metabolomics analysis and found that nucleotide-related metabolites were more abundant in SKOV3/DDP. The enrichment analysis of the different metabolites is even more indicative of purine metabolism. This suggests that purine metabolism is a likely future therapeutic target for platinum resistance in OC. We will explore this direction in more detail in the future.

Undeniably, there are some limitations of this signature at present. First, this signature has not been verified by expansion in the reality database, and its diagnostic validity still needs further confirmation. Whether there will be variability among different ethnic groups is unknown. Second, we currently do not have access to mutational profiles in these patients due to the lack of public data on recurrent or drug-resistant ovarian cancer. We will try to explore this in the future, which will be important to enhance ovarian cancer diagnosis and treatment. Finally, we propose potential therapeutic approaches that are primarily based on bioinformatics. Precision therapy is extremely attractive and challenging. A large number of experiments are needed in the future to validate their efficacy in precision therapy.

Overall, our prognostic signature based on nine PRGs was able to accurately distinguish between high- and low-risk subgroups and performed well as an independent prognostic factor in predicting the survival of OC patients. Potential clinical drugs were then screened for the PRG-based prognostic signature based on immune infiltration and drug sensitivity analysis to facilitate decision-making and clinical practice. Meanwhile, a comprehensive nomogram with a combination of PRG-based prognostic signature and other clinical features was constructed to further effectively predict the prognosis of OC patients. Our study provides new insights into prognostic prediction and treatment of OC and provides an important basis for future studies of purine metabolism or PRGs in OC.

## Figures and Tables

**Figure 1 jpm-13-00776-f001:**
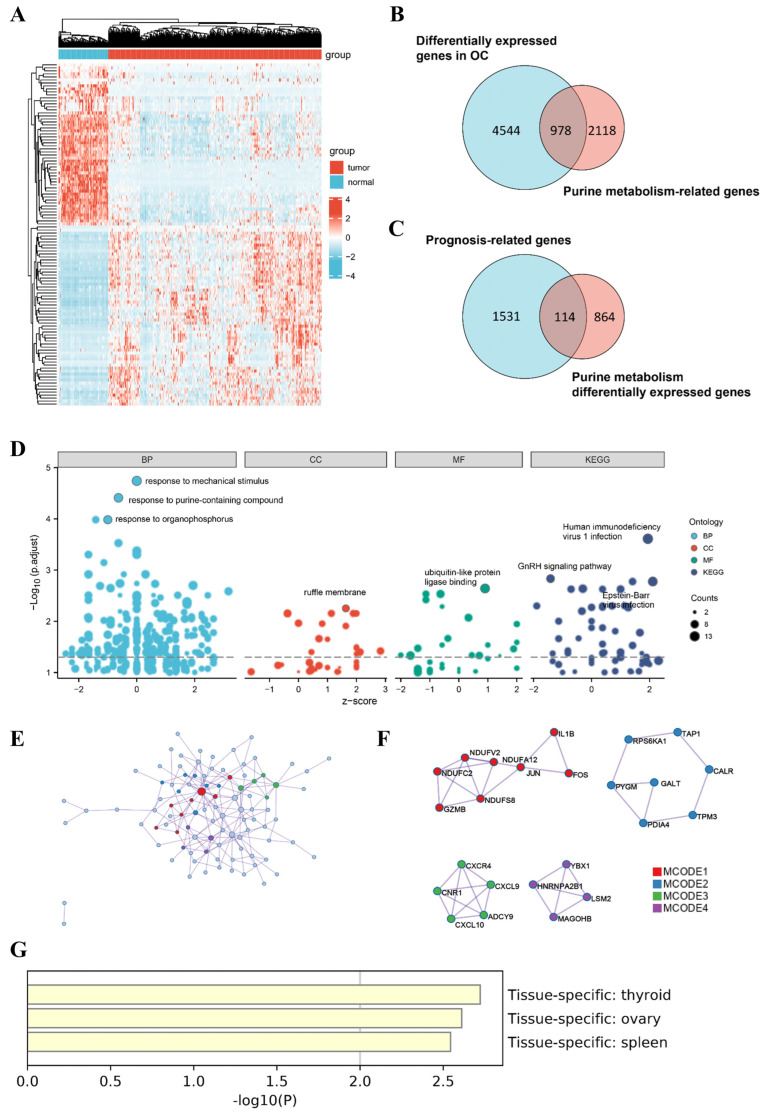
Identification of dysregulated PRGs. (**A**) Heatmap of DEGs in OC and normal ovarian tissue from TCGA. (**B**) Screening for differentially expressed PRGs and (**C**) correlation with prognosis. (**D**) Enrichment analysis of 114 PRGs based on GO and KEGG analysis. (**E**) PPI network and (**F**) MCODE showing hub genes. (**G**) Tissue-specific enrichment analysis based on PaGenBase. DEG, differentially expressed gene. OC, ovarian cancer. TCGA, The Cancer Genome Atlas. PRG, purine-metabolism-related gene. PPI, protein–protein interaction. MCODE, molecular complex detection.

**Figure 2 jpm-13-00776-f002:**
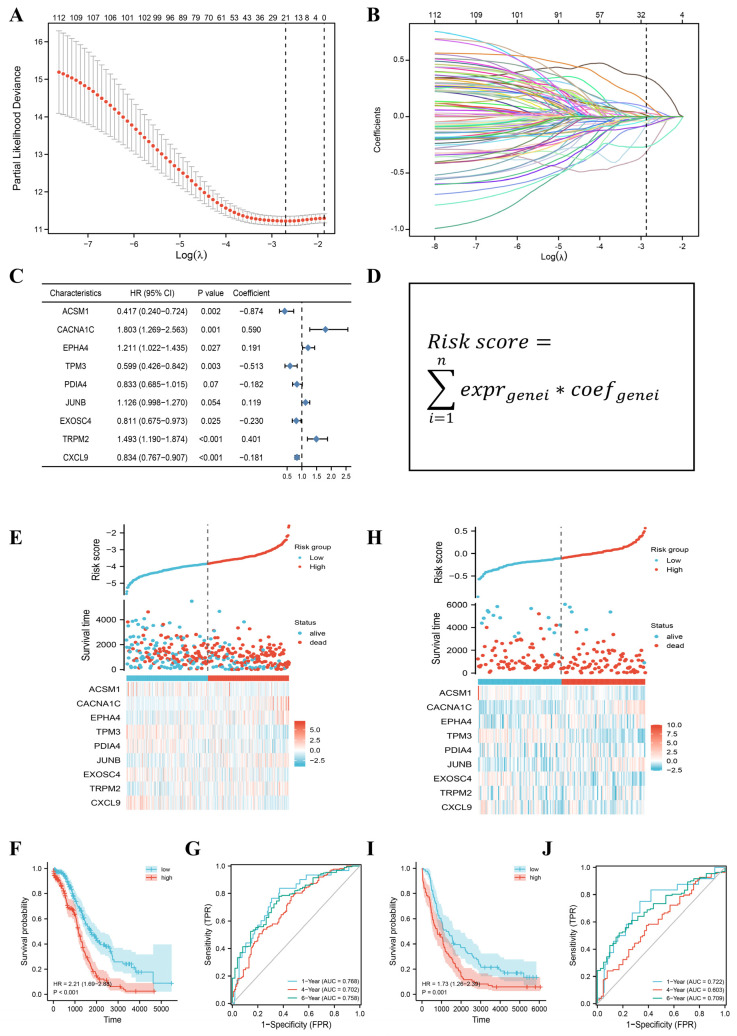
Construction and validation of a prognostic signature based on 9 PEGs. (**A**) LASSO regression analysis of 114 PRGs. (**B**) LASSO variable trace plots. The different colored lines represent a wide range of candidate genes. (**C**) Multivariate cox regression for 9 PRGs. (**D**) Risk score formula. (**E**) Distribution, survival status, and gene expression profiles of OC patients based on median risk scores in the TCGA cohort. (**F**) Kaplan–Meier curve in the TCGA cohort. (**G**) ROC curves for 9 PRGs in the TCGA cohort. (**H**) Distribution, survival status, and gene expression profiles of OC patients based on median risk scores in the GEO cohort. (**I**) Kaplan–Meier curve in the GEO cohort. (**J**) ROC curves for 9 PRGs in the GEO cohort. GEO, Gene Expression Omnibus. HR, hazard ratio. ROC, receiver-operating characteristic.

**Figure 3 jpm-13-00776-f003:**
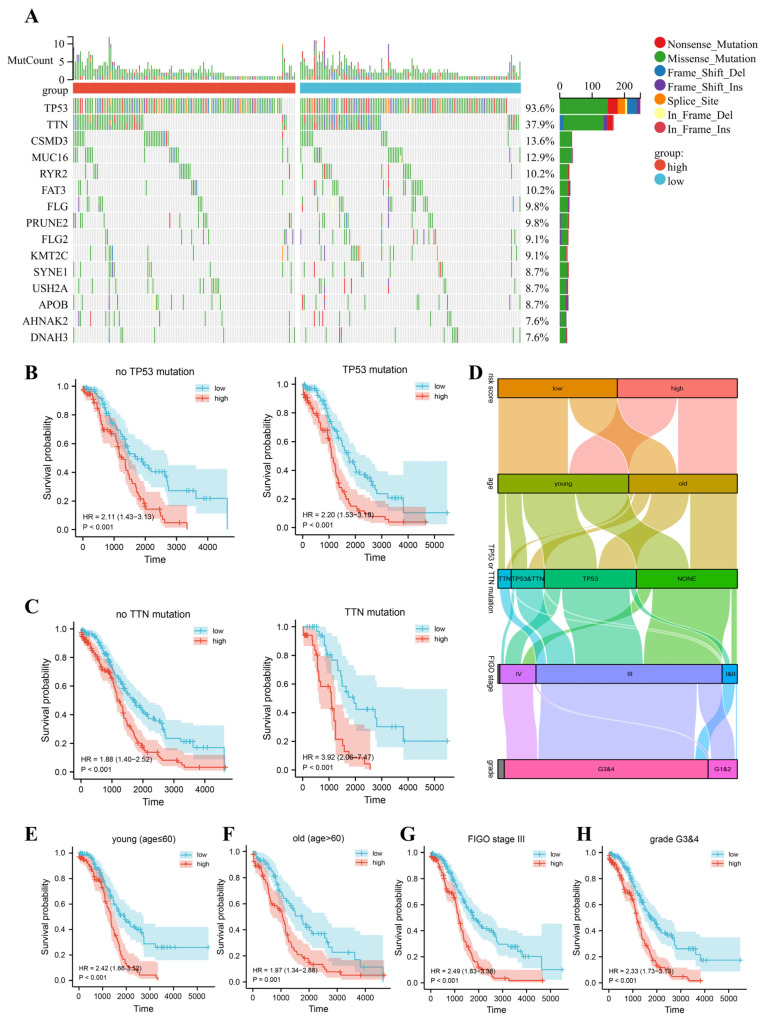
Survival prognosis of different mutation subtypes and different clinical subgroups. (**A**) Mutational landscape in the TCGA cohort. (**B**) Kaplan–Meier curves for TCGA cohorts with or without TP53 mutations. (**C**) Kaplan–Meier curves for TCGA cohorts with or without TTN mutations. (**D**) Risk score levels, mutations, and clinical features of patients in the TCGA cohort. (**E**–**H**) Kaplan–Meier curves based on different clinical features in the TCGA cohort.

**Figure 4 jpm-13-00776-f004:**
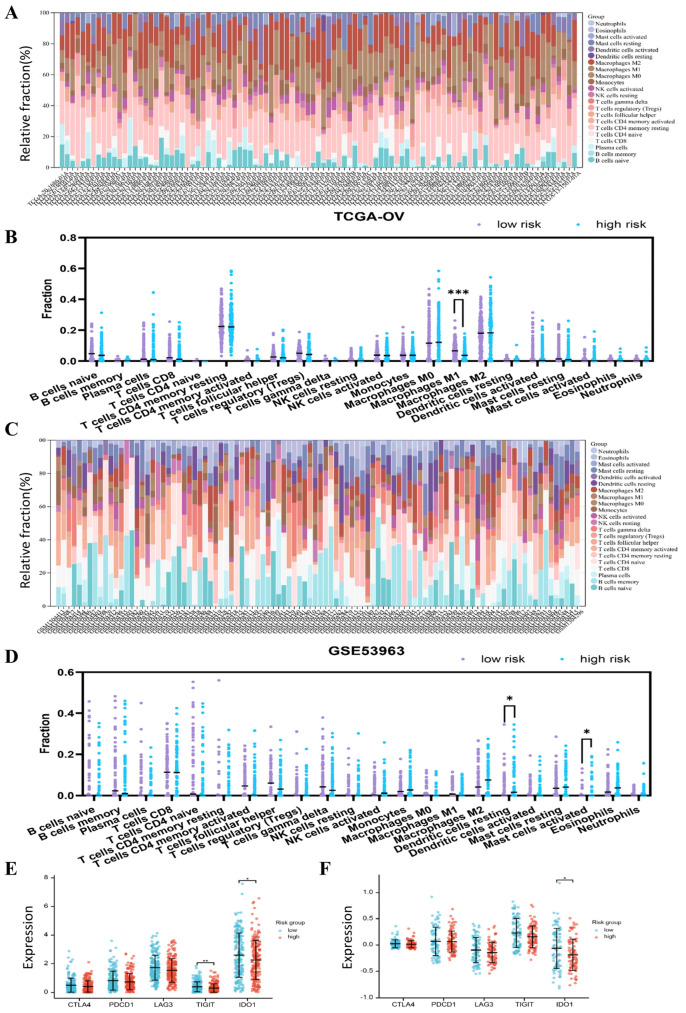
Immune cell landscape and immune checkpoints in different risk subgroups (*, *p* < 0.05; **, *p* < 0.01; ***, *p* < 0.001). (**A**) Immune cell landscape in the TCGA cohort. (**B**) Proportion of infiltrating immune cell subsets according to risk subgroup in the TCGA cohort. (**C**) Immune cell landscape in the GEO cohort. (**D**) Proportion of infiltrating immune cell subsets according to risk subgroup in the GEO cohort. (**E**) Immune checkpoint expression in the TCGA cohort. (**F**) Immune checkpoint expression in the GEO cohort.

**Figure 5 jpm-13-00776-f005:**
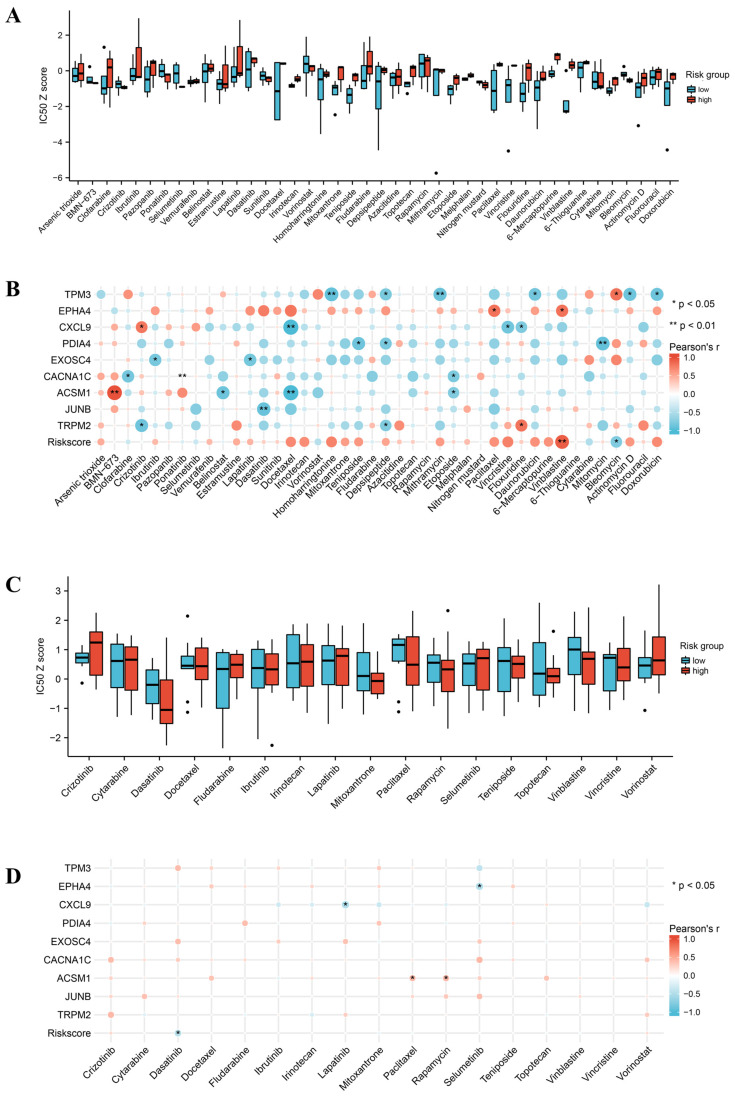
Association of risk subgroups or scores with drug sensitivity (*, *p* < 0.05; **, *p* < 0.01). (**A**) Drug sensitivities of different risk subgroups were obtained from CellMiner. (**B**) Correlation of drug IC50 with expression levels of 9 PRGs and risk scores in CellMiner. (**C**) Drug sensitivities of different risk subgroups obtained from GDSCv2. (**D**) Correlation of drug IC50 with expression levels of 9 PRGs and risk scores in GDSCv2. IC50, drug half-inhibitory concentration.

**Figure 6 jpm-13-00776-f006:**
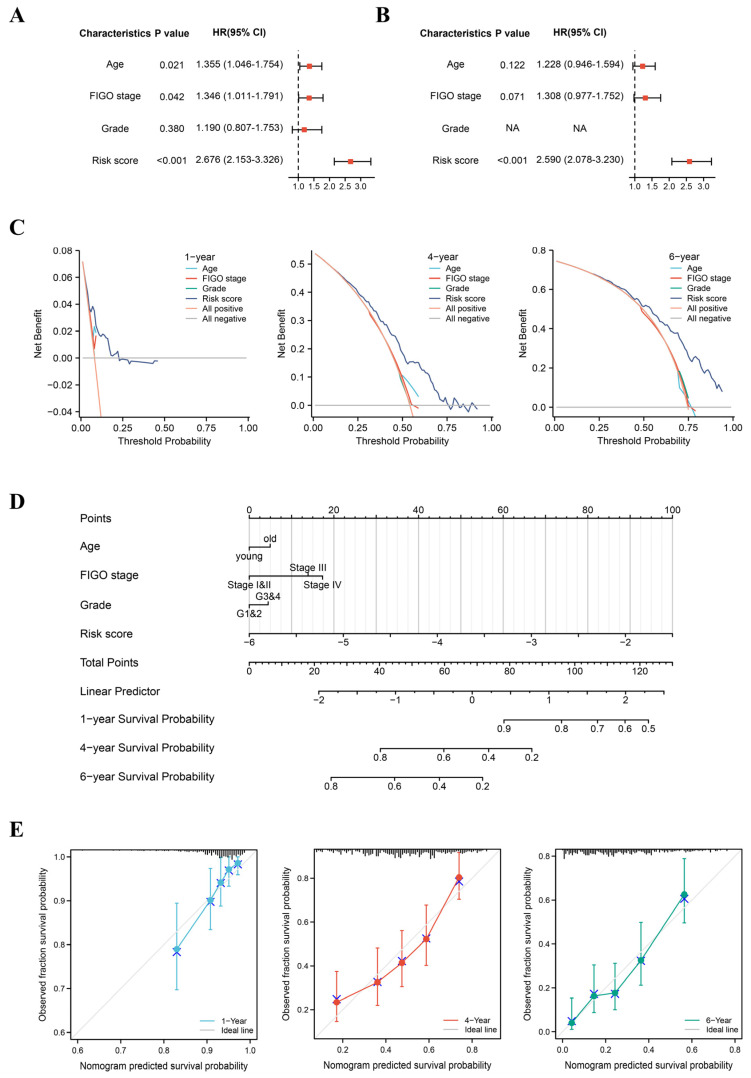
Construction of the nomogram. (**A**) Univariate Cox regression analysis of clinical features and risk scores in the TCGA cohort. (**B**) Multivariate Cox regression analysis of clinical features and risk scores in the TCGA cohort. (**C**) Decision curve analysis plots. (**D**) Nomogram for predicting 1-year, 4-year, and 6-year OS in OC. (**E**) Calibration curves for nomogram. OS, overall survival.

**Figure 7 jpm-13-00776-f007:**
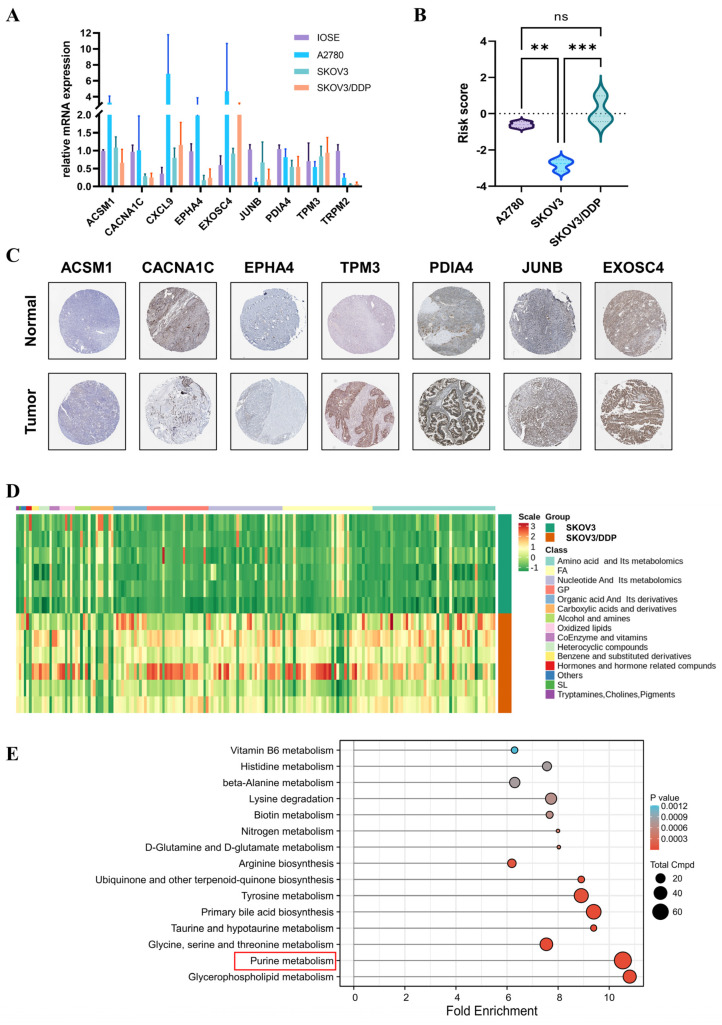
The mRNA and protein expression levels of nine PRGs(**, *p* < 0.01; ***, *p* < 0.001). (**A**) The mRNA expression levels of 9 PRGs in IOSE, A2780, SKOV3, and SKOV3/DDP. (**B**) Risk scores of A2780, SKOV3, and SKOV3/DDP. (**C**) Immunohistochemical data from the HPA database. (**D**) Heatmap of the comprehensive nontarget metabolomic analysis of SKOV3 and SKOV3/DDP. (**E**) KEGG enrichment of differential metabolites between SKOV3 and SKOV3/DDP. The red frame marks the purine metabolism pathway.

**Table 1 jpm-13-00776-t001:** Univariate and multivariate analysis for 9 PRGs in the TCGA cohort.

Gene	Univariate Analysis			Multivariate Analysis		
	HR	95% CI	*p* Value	HR	95% CI	*p* Value
*ACSM1*	0.730	0.563–0.947	0.018	0.687	0.525–0.898	0.006
*CACNA1C*	1.390	1.073–1.800	0.013	1.187	0.900–1.565	0.225
*EPHA4*	1.352	1.042–1.753	0.023	1.341	1.022–1.761	0.034
*TPM3*	0.647	0.499–0.839	0.001	0.707	0.538–0.929	0.013
*PDIA4*	0.698	0.538–0.905	0.007	0.698	0.531–0.917	0.010
*JUNB*	1.552	1.195–2.017	<0.001	1.312	0.992–1.735	0.057
*EXOSC4*	0.685	0.528–0.888	0.004	0.689	0.525–0.903	0.007
*TRPM2*	1.399	1.077–1.816	0.012	1.412	1.065–1.871	0.017
*CXCL9*	0.713	0.549–0.927	0.011	0.672	0.512–0.883	0.004

## Data Availability

The data used in this study were obtained from published reports, and there is no need to provide an additional statement of permission/consent for these databases. The data used to support the findings of this study are available from the corresponding author upon request.

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
