# Peer review of "Comprehensive Analysis of Purine-Metabolism-Related Gene Signature for Predicting Ovarian Cancer Prognosis, Immune Landscape, and Potential Treatment Options"

_jpm, 2023, doi:10.3390/jpm13050776_

Round 1

Reviewer 1 Report

Dear Authors,

Thank you for an very interesting paper. Future investigations are need to improve management of OC but research like this makes it promising. 

Have you investigated mutations in patients operated due to the relapse of the disease? It would be interesting to have these data as well.

Thank you,

Reviewer

Reviewer 2 Report

Dear Editor,

First of all, thnak you for giving this opportunity to me.

I have read this interesting article which is aimed to investigate some gene profiles of the ovarian caner.

As we know that the TGCA have been implemented in the field of endometrial cancer. So ovarian cancer research are becoming more important in the literature.

The scientific language of the study is fine. The methodology is ok.

the results are well documented and the figures are quite enough.,

It can be published in this journal.

Author Response

We sincerely appreciate your careful and insightful review.

We hope that this manuscript will help to personalize the diagnosis and treatment of ovarian cancer in the future.

Reviewer 3 Report

The manuscript by Liu et al successfully develops a purine metabolism-based gene expression profile for use in ovarian cancer prognosis. The PRG low/high risk profile matches up astonishingly well with survival curves regardless of mutational status and may be a valuable prognostic tool moving forward.

Individual notes/comments:

1. The authors note that the prognosis of the high-risk group was worse regardless of the presence of a TTN mutation, but the hazard ratio with a TTN mutation is more than double than without a TTN mutation. The authors should note that their expression profile may be of even greater prognostic use in patients with TTN mutations.

2. It is very difficult to see the line marking the mean in figure 4B and D – perhaps make the line black instead of purple/blue so that readers can appropriately judge the data for themselves.

3. While the IDO1 expression is significant by statistically tests this appears to be more due to low variability than remarkably different expression levels. The statement that IDO1 inhibitors would be of more benefit to the low risk group is likely overstated. 

4. The author claims that CellMiner and GDSCv2 trends are “nearly consistent”, yet the same drugs do not pop up as significant in both sets. The consistency is more in the fact that many drugs do not show a correlation with risk score. The CellMiner dataset looks good but only indicates bleomycin and vinblastine as correlated with risk score, and vinblastine is not consistent with the GDSCv2 data. I think the authors should omit their claim that dastanib has a correlation since the data appear to be at odds with one another. TheIC50 is lower in low risk than high risk in GDSCv2 and the opposite in the CellMiner data

Reviewer 4 Report

In my opinion, the analyzed topic is interesting enough to attract the readers’ attention. The goal of this  article was to achieve one of the first comprehensive analysis of genes related to purine metabolism in ovarian cancer patients and create a feasible prognostic signature that will aid in risk prediction and support personalized medicine.

I think that the abstract of this article is well organized and clear.

In my opinion, the discussion could be studied in depth and extended. Maybe, it could be useful the evaluation ,as a comparison, of recent insights of management of gynecological cancers also on molecular aspects. In particular I suggest these two articles  PMID: 36979434 and PMID: 36983243. Because of these reasons, the article should be revised and completed. Figures and tables are clear. Considered all these points, I think it could be of interest for the readers and, in my opinion, it deserves the priority to be published after minor revisions.

Reviewer 5 Report

I was pleased to review this paper.

The methodology used by the Authors is appropriate for the purpose of the article description. The English language is fluid and well understood. Nevertheless, minor revision is needed.

For the abstract I found clear  the schematization of the content.

Regarding the introduction you can add this paper to deepen the topic.

DOI:  10.21037/gs-20-544.

There is a good structure of the project: I find this work project well structured.

This analysis of genes related to purine metabolism in ovarian cancer patients and created a feasible prognostic signature that will aid in risk prediction and support personalized medicine: What do you think could change in the future?

What errors could one stumble into from using this analysis?

When instead you absolutely would not recommend to consider your model?

 Is there more difficulty in analyzing drug resistant cellular systems?

The statistical structure seems to me well done;

Good subdivision into subparagraphs;

Mention the limitations of the study;

The data obtained are good with respect to the structuring of the work;
